

# Multi-Scale Virtual Field Experience, Grand Ledge, Michigan, USA

Madeline S. Marshall[1], Melinda C. Higley[2]

[1]Department of Earth & Environment, Albion College, Albion, Michigan, 49224, USA
[2]Geology, Geography, and Environmental Studies Department, Calvin University, Grand Rapids, Michigan, 49546, USA

*Correspondence to*: Madeline S. Marshall (mmarshall@albion.edu)

**Abstract**. Field experiences are a critical component of undergraduate geoscience education; however, traditional onsite field experiences are not always practical due to accessibility, and the popularity of alternative modes of learning in higher education is increasing. One way to support student access to field experiences is through virtual field trips, implemented either independently or in conjunction with in-person field trips. We created a virtual field trip (VFT) to Grand Ledge, a regionally important suite of

sedimentary outcrops in central lower Michigan, USA. This VFT undertakes all stages of a field project, from question development and detailed observation through data collection to interpretation. The VFT was implemented in undergraduate Sedimentation and Stratigraphy courses at two different liberal arts institutions, with one version of the VFT conducted in-person and the other online. The VFT was presented from a locally hosted website and distributed through an online learning platform. Students completed a series of activities using field data in the form of outcrop photos, virtual 3D models of outcrops and hand

samples, and photos of thin sections. Student products included annotated field notes, a stratigraphic column, a collaborative stratigraphic correlation, and a final written reflection. VFT assessment demonstrated that students successfully achieved the inquiry-oriented student learning outcomes and student reflection responses provide anecdotal evidence that the field experience was comparable to field geology onsite. This VFT is an example of successful student learning in an upper-level Sedimentation and Stratigraphy course via virtual field experience with an emphasis on local geology.

**1 Introduction and motivations**

Field experiences for undergraduate geoscience students are key exercises in which students integrate classroom knowledge with real-world examples, implement skills, gain vocational experience and insight, and practice collaborating with a field team (Mogk and Goodwin, 2012; Petcovic et al., 2014). Bringing the field experience to students, rather than always taking students into the field, is increasingly important for reasons of accessibility, time, cost, and offering comparable opportunities to online students

(Huntoon, 2012; Arthurs, 2021; Rotzien et al., 2021). Despite a long history of field trips in the geosciences, some desktop-based virtual field trips have, in fact, been shown to yield better learning experiences and outcomes than actual field trips (Zhao et al., 2020).

We both teach undergraduate-level Sedimentation and Stratigraphy courses that typically include a field trip, often to Grand Ledge,

the closest major suite of sedimentary outcrops in Michigan with a variety of lithologies and fossils (Kelly, 1933; Martin, 1982). Our goal in the development of this VFT was to create an accessible and remote field experience that undertakes all stages of a field project, from question development through data collection to interpretation. In the implementation of this VFT, the Albion College course was entirely online, and the Calvin University course was in-person.

Albion College is a comprehensive liberal arts institution with an enrollment of ~1500 undergraduate students. Albion prioritizes building a culture of belonging and experiential learning, and preparing students to translate critical thought into meaningful action.





Calvin University is a comprehensive liberal arts college with an enrollment of ~3000 undergraduate students. Learning at Calvin is rooted in its Christian Reformed commitment and in the study of geosciences at Calvin, we pursue intellectual efforts to explore our world's beauty and engage in stewardship of Earth's resources.


The objectives of this project included: (1) giving students an opportunity to explore outcrops in detail, which is valuable as an independent virtual experience or as preparatory work for going out in the field ; (2) creating a structure that would be expandable and ongoing in its scope, with our future goals being to incorporate subsurface data and samples from the Michigan Core Repository; (3) addressing issues of accessibility, disorientation, limited data, and inflexible scope, which can make some VFT

experiences less wholistic and satisfying than in-person field trips (e.g., Hall et al., 2004; Carabajal et al., 2017); (4) a broader goal of this Grand Ledge VFT was to thoroughly document and encourage the preservation of a suite of historically and geologically important Pennsylvanian outcrops in Grand Ledge, Michigan (e.g., Milstein, 1987a).

For each section that follows, we describe part of the VFT Assignment (see Supplement), its importance and our expectations,

discuss how it was implemented, and evaluate the outcomes (including student examples where relevant). The complete assignment, rubric, and other materials are available in the Supplement, and further information about the VFT and materials are available at

https://serc.carleton.edu/NAGTWorkshops/online_field/activities/242310.html.

### 1.1 Student learning outcomes (SLO)

In this VFT, students focused on skill development, instead of working towards a single "right" answer. The dual emphases of the VFT were for students to develop conceptual understanding (e.g., recognizing and describing bedding styles) and research skills (e.g., hypothesis testing and analysis). Upon successfully completing this project, students will be able to:

1. Apply their course knowledge to analyze the stratigraphic characteristics of a real-world field site through a virtual field experience
2. Identify and describe lithologies from a combination of outcrop photos, 3D models, and thin sections.
   3. Recognize and describe bedding styles and geometry from outcrop photos and 3D models.
   4. Create a detailed, (litho)stratigraphic column using data from SLOs 2-3 and additional stratigraphic column resources.
   5. Develop an interpretation of the depositional environment(s) for the stratigraphic column.
6. Present final products and discuss observations and the strengths and weaknesses of different interpretations.

### 1.2 Deliverables

Through the course of the VFT, students produced three major deliverables. First, they submitted a copy of their field notes, including an annotated outcrop sketch, rock descriptions, and their original paleoenvironmental hypothesis (Parts 2-4 of the Assignment; see Supplement). Second, each student submitted an original stratigraphic column with their graphical log, descriptive

notes, and interpretations for each unit (Part 5 of the Assignment). Third, after completing all other VFT components and engaging in a group discussion of the stratigraphic correlations, as well as discussing supporting literature, students wrote a final reflection (Part 7 of the Assignment). In our experiences, students were able to complete most VFT components during allotted class and lab time: 10 in-person hours for Calvin students, 12 online hours for Albion students. This dedicated in-class time over the course of 4-5 total days was essential to engage in the group work, particularly when considering that collaboration and discussion on an





outcrop is a key part of in-person field work. In assessing student learning in this VFT, all students have unique products tied to different primary outcrops, making plagiarism a non-issue even in a virtual setting. Thus, rubrics were designed to account for the inherent variability of student products, and focused largely on skill development.

### 1.3 VFT development

Over the course of a month, we invested substantial time to prepare this virtual experience, which included two full days of
fieldwork, plus several weeks of image and data processing. When collecting field data to prepare this VFT, it was essential to document outcrops through photos and virtual 3D models at many scales and from multiple angles. This resulted in an experience for students similar to walking around all sides of an outcrop, stepping back, and moving in closer. In the development phase of the VFT we documented each outcrop through the following methods:

1. Explored the outcrop and marked several important places on the outcrop to be used for close-up photos. Previous work
85        at Grand Ledge guided our outcrop choices (e.g., Martin, 1982; Venable et al., 2013).

2. 360° photos of each outcrop were taken with a GoPro Max 360° camera. The 360° photos provided a sense of orientation for the outcrop and surrounding landscape and context.

3. Photos of the outcrop were taken at multiple levels in approximately the same plane and were compiled into an outcrop panorama photo using Adobe Photoshop (Fig. 1). Of note, photos taken using an iPhone overcame the issue of dappled
90        shadows on the outcrops significantly better than other cameras we used.

4. Additional photos of the outcrop were taken at multiple levels and from multiple angles and were compiled into a high-resolution 3D outcrop model using Agisoft Metashape Professional (2020) software and then uploaded to Sketchfab, where the marked locations of close-ups were added as annotations to the model (Fig. 2).

5. Close-up photos were taken of each marked feature. These included examples of bedding for each geologic unit, fossils,
95        and examples of the varieties of sedimentary structures in the outcrop (Fig. 3).

6. We collected small, loose hand samples from several close-up areas that would benefit from further imaging or thin sectioning.

7. Photos of each hand sample were taken using a turntable and were processed into a 3D hand sample model using Agisoft Metashape Professional (2020) software and then uploaded to Sketchfab (Fig. 4).

8. We prepared 14 of the hand samples into thin sections, resulting in 1-3 representative samples from each locality, to capture the lithologic variation through the stratigraphy in each region of the VFT. These thin sections were photographed under plane polarized and cross-polarized light at multiple magnifications, to provide a suite of photomicrographs for students to evaluate thin sections in a virtual setting (Fig. 5).





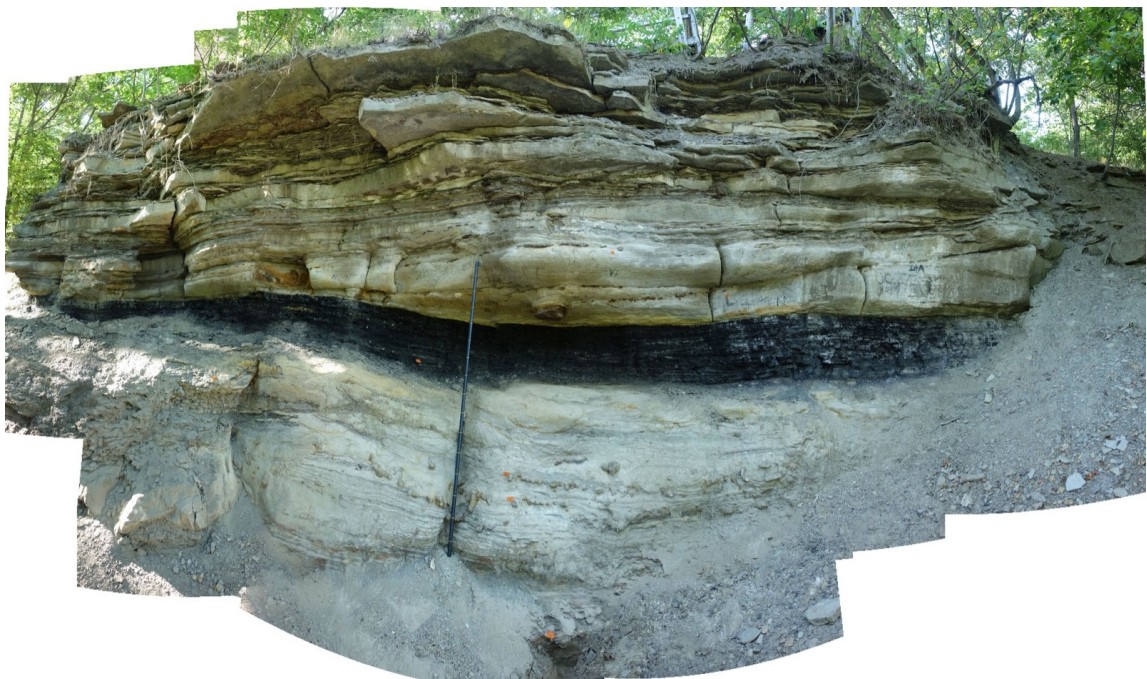

**Figure 1:** Stitched panorama photo of American Vitrified 2 outcrop (staff is 1.6 m and has 10 cm increments). This image is used in Parts 2-5 of the Assignment. Figure 2 is the equivalent 3D model of this outcrop, and the corresponding stratigraphic log is in Fig. 6.

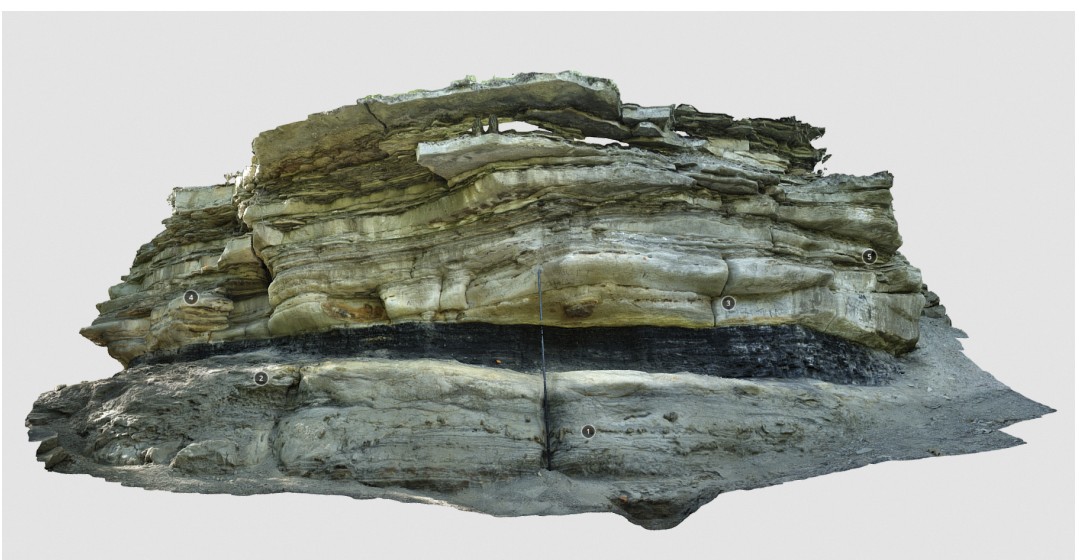

**Figure 2:** Virtual 3D outcrop model of American Vitrified 2 (staff has 10 cm increments); annotation numbers correspond to other photo and sample data in the VFT. This is a still image of the fully manipulatable 3D virtual model of the outcrop, available via Sketchfab. This model is used in Parts 2-5 of the Assignment.



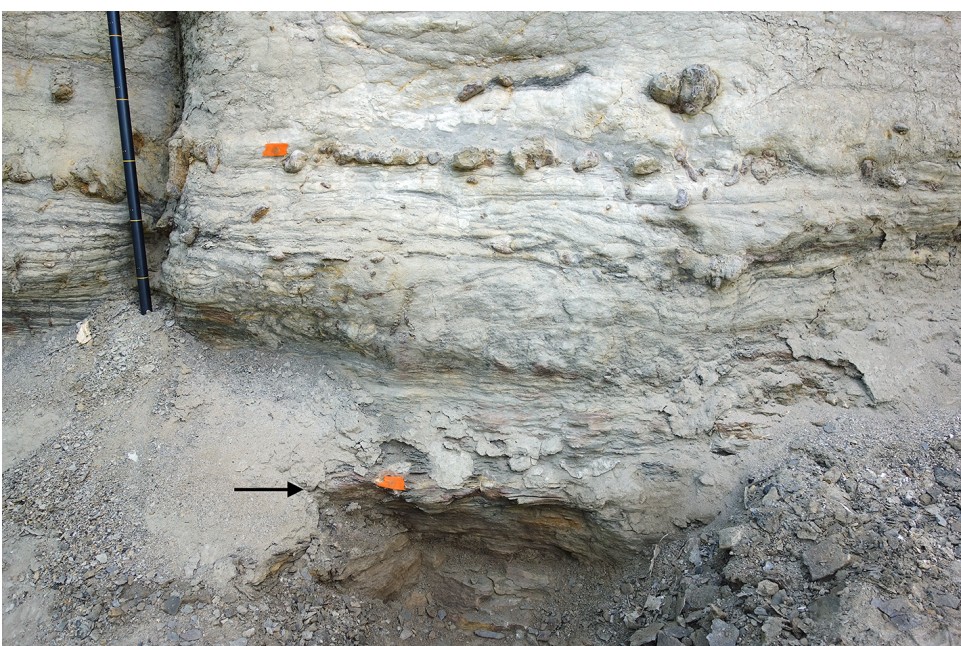

**Figure 3:** Close-up photo of the base of the American Vitrified 2 outcrop, with location of the basal mudstone sample (AV 2-1 in Fig. 4) shown with the black arrow and orange tape (staff has 10 cm increments and is in the same location in Figs. 1-3). This photo is used in Parts 3-5 of the Assignment.

**(a)**                                      **(b)**

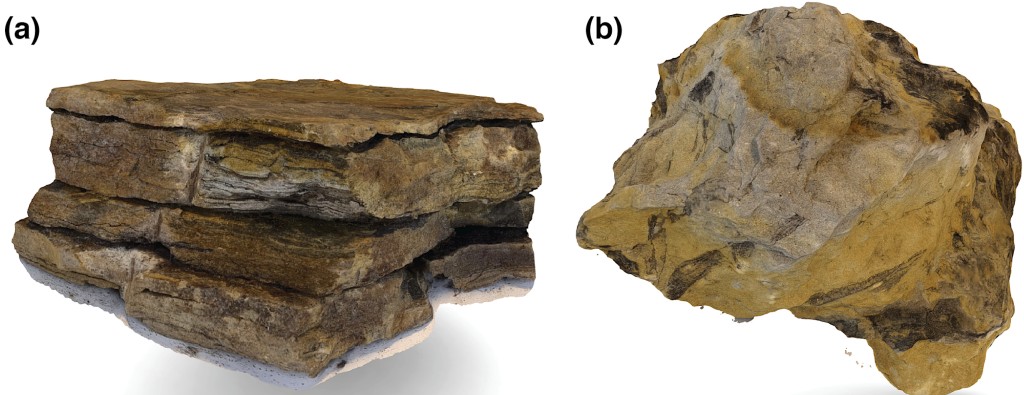

**Figure 4:** 3D virtual models of hand samples from American Vitrified 2 outcrop: (A) AV 2-1 mudstone from the base of the outcrop (arrow in Fig. 3), maximum diameter of sample is 13 cm; and (B) AV 2-2 sandstone with *Stigmaria* from the outcrop model annotation 2 (in Fig. 2), maximum diameter of sample is 15 cm. These samples are available as models AV2-1 and AV2-2 via Sketchfab; scale bars are provided separately in still photos of these hand samples hosted for students on the project website.



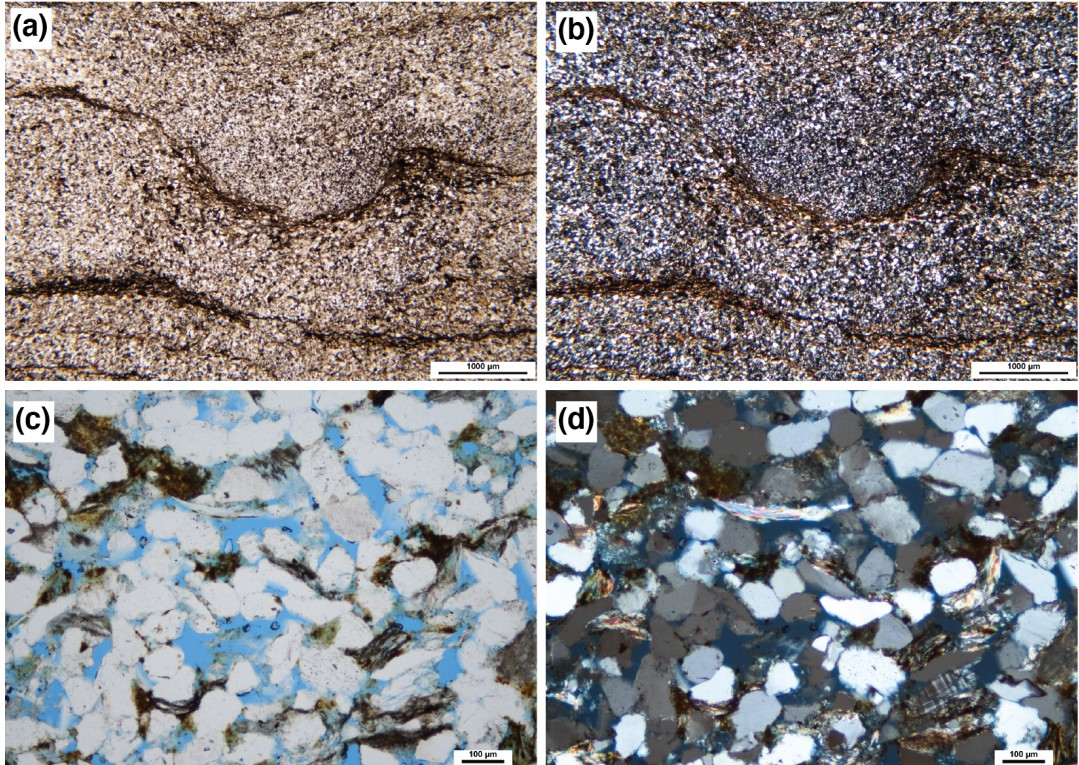

**Figure 5:** Thin section images from American Vitrified 2, used in Part 3 of Assignment for students to precisely determine lithologies. (A) Sample AV 2-1 in PPL, the organic-rich mudstone from Fig. 4, (B) sample AV 2-1 in XPL from Fig. 4, (C) sample AV 2-2 in PPL, the sandstone from Fig. 4, and (D) sample AV 2-2 in XPL from Fig. 4. Scale bars in A-B are 1000 μm, and scale bars in C-D are 100 μm.

**1.4 Ethics**

This study describes a virtual field experience project completed in our respective classrooms in the fall of 2020. This was not a research project involving human subjects, students were not surveyed, and we do not report data about human subjects. Written permission was obtained from students whose work is presented here as examples.

**2 Background geology and framework: Part 1 of assignment**

In class periods leading up to the VFT, students engaged with concepts of lateral and vertical facies relationships, drivers of sea-level change, and environments of deposition. The first part of the VFT was designed to orient students to time and place, establishing background geology of the Paleozoic of the Michigan Basin and making connections to broader sedimentology and stratigraphy topics (preliminary work on SLO #1). The students used Google Earth imagery at multiple scales, both statewide and focused on their unique outcrops in Grand Ledge, Michigan. Students worked through provided written and graphical information about the Late Paleozoic structure, stratigraphy, and climate of the Michigan Basin and surrounding areas (Milstein, 1987b; Catacosinos et al., 2000; Haq et al., 2008; Towne et al., 2013; Venable et al., 2013). They also discussed the following two sets of questions with their small group, coordinating theory with the evidence presented, and recording their hypotheses in their field notes: (1) Why was relative sea level low during the early Pennsylvanian? Hypothesize about the different drivers of relative sea level change that may have caused this. How might those drivers have also influenced the sedimentary record we will observe? (2)





What environmental changes can you recognize in the Michigan Basin stratigraphy of the late Paleozoic? Are there particular times of high or low relative sea level?

Importantly, students were tasked with developing preliminary hypotheses about the Pennsylvanian depositional environments of Grand Ledge, which they would subsequently revise. This step of the VFT introduced important vocabulary and context, and was

structured to connect with concepts discussed in class. This step of fieldwork asks us to establish familiarity with when, where, and what is above and below the interval of interest, so that the context can better inform our hypotheses and interpretations.

### 2.1 Discussion of background and framework

Establishing the context for the virtual field experience, including some regional geology and sea-level history, was a critical part of students understanding why Grand Ledge was an interesting locality to study and visit virtually. Student response to the

background section was mixed, and this is one aspect that could have benefited from an initial class discussion. On in-person field trips, the leader or instructor often gives the background context before participants head to the outcrops, and a similar approach could be valuable herein. However, the goal of the background section of the project was also for students to gain more experience reading maps, stratigraphic columns, and sea-level curves, and formulating hypotheses based on these. To that end, a recommended improvement would be to require students to submit these initial hypotheses immediately, or have the instructor validate each

hypothesis. Ideally, this part of the VFT should expand their field notes to address the context around questions 1 and 2 above: time periods of interest, primary depositional environments, paleoclimate, and field site location information.

### 3 Outcrop reconnaissance:  Part 2 of assignment

The second part of the VFT was to become oriented to the outcrop and make an annotated sketch to record initial observations, similar to how one approaches a new outcrop in person by first taking in the big picture (preliminary work on SLO #1). The

students used Google Earth imagery, 360º photos, and began exploring all of the outcrop and hand sample virtual 3D models and photos for their site (Figs. 1-4). In small groups of 2-3, students first accessed the materials for their assigned field site to conduct reconnaissance through a cursory examination of the available maps, imagery, and descriptions. Students then used the 3D outcrop model and outcrop photos to make an outcrop sketch in their notes; on their sketch, each student defined distinct lithologic units and annotated their sketch with any observations they could make. Students were also encouraged to record preliminary questions

and hypotheses to drive their subsequent investigation. This step of the VFT yielded the primary product of an outcrop sketch with labeled units and annotated features, accompanied by notes on preliminary hypotheses about the lithologies and paleoenvironmental interpretations. This aspect of fieldwork is essential, and was incorporated to make sure that students could establish their orientation and understanding of the site in space, which is a challenge both in person and virtually, and is a significant accessibility issue (Hall et al., 2004). Utilizing virtual outcrop models has also been shown to be a positive and effective

experience for students when engaging in virtual field experiences and developing 3D spatial thinking skills (Bond and Cawood, this volume).

### 3.1 Discussion of outcrop reconnaissance

Prior to the virtual field trip, we spent time scaffolding by discussing principles of making a good outcrop sketch, practicing as a class, and assessing examples (Geology Drawing Skills Handbook, 2018). That preparation made a difference in the confidence

and skill of students as they approached the VFT data, given the particular challenge of sketching an outcrop with only virtual





materials. One of the challenges for students was to focus on the original sedimentary features, looking past rubble, fractures or joints, weathering stains, and vegetation. While most students depicted accurate shapes and beds for their outcrops, sketches varied from being highly stylized to largely realistic.

## 4 Lithologies: Part 3 of assignment

The third part of the VFT was to identify and describe lithologies from a combination of outcrop photos, 3D models, and thin sections (SLO #2). The students used close-up photos tied to annotations on the 3D outcrop models (Fig. 2), 3D hand sample models and photos (Fig. 4), and thin sections made from the hand samples (Fig. 5). Notes accompanied some of the imagery to clarify locations and relationships that would aid in understanding orientations. Students first differentiated distinct units within their outcrop and then wrote a rock description for each unit. Students then studied thin sections and completed ternary QFR

diagrams (Folk, 1980) with the goal of refining their rock descriptions. This step of the VFT yielded products of rock names and their QFR constituent percentages, textural descriptions, and hypotheses of depositional environments for each lithologic unit described. Additionally, students experienced the complexities and challenges of making lithologic observations in the field.

### 4.1 Discussion of lithologies

To accurately describe lithologies, it is important to examine multiple scales of data. In the VFT, we achieved this through outcrop

photos, 3D models and photos of hand samples, and photomicrographs – this approach also replicated the order of in-person fieldwork and subsequent laboratory investigations.

The outcrop is the first point of interaction in the field experience, requiring broad scale observations. In asking students to define their own units in each outcrop, it rapidly became clear who was looking closely at details and who was rushing to finish: students

who had not examined all parts of their available data during the reconnaissance stage invariably noted fewer units than there were hand samples for. This led to revision of their hypotheses as they collected more data at finer scales.

In the field, it is often more intuitive for students to zoom in and out of an outcrop, stepping closer to view more details. In the VFT, students had to learn that the discrete images and models at different scales were the equivalent experience. Photos of hand

samples and 3D models are high enough resolution that rock texture is visible therefore students can make observations about grain size and mineralogy. Students did need instructor assistance to determine exactly what information could be gleaned from a hand sample. This is similar to our experiences with students interacting with physical samples in the field or lab, with particular emphasis on encouraging students to look at the primary structures and lithology, as opposed to weathering rinds or fractures.

Thin section data was a valuable component of the lithology section. At the Grand Ledge outcrops, the differences between some of the rock units are subtle (e.g., silt versus very fine sand), and grain size information is key to distinguishing them. Thin section photomicrographs allowed students to test their hypotheses of grain size, which until this point in the VFT were based on hand sample data and the general appearance of the outcrop. Students estimated percent distributions of minerals and plotted this data on QFR diagrams. Incorporating a semiquantitative component was important in offering students a concrete way to back up their

interpretations. With thin section data students are challenged to revise estimates of grain size, evaluate sediment provenance, and consider possible depositional environments.





By presenting highly organized data in the VFT, it seemed possible that students would approach the VFT as if the instructors had prescribed the answers. However, students approached the VFT data with fresh eyes, and did not necessarily follow the units or order of operations we had planned. This was a surprising and positive outcome, and showed that a VFT can be a genuine exploration for students.

## 5 Bedding style and sedimentary structures: Part 4 of assignment

The fourth part of the VFT focused on recognizing and describing bedding styles and geometry from outcrop photos and 3D models (SLO #3). Students evaluated outcrop photos and 3D outcrop models (Figs. 1-3) to measure bed thicknesses in their outcrops and assess trends through the section (i.e., thinning or thickening up-section). They next identified and carefully described any sedimentary structures (e.g., flaser bedding, trough cross-bedding, burrows) and clasts (e.g., rip-up clasts, fossils of plant material) they saw, which were often difficult to distinguish due to the nature of the outcrops. Both of these components were added to their outcrop sketches and notes to refine their initial work. Students then revised their environmental interpretation hypotheses based on this new data, and were encouraged to focus on how the energy, sediment supply and type, and life present changed over time, through the succession of units present. This step of the VFT yielded products of actual measurements of bed thicknesses and named sedimentary structures for their outcrop, which required students to step back from the lithologic details and reassess the stratigraphic patterns. This step presents the challenge of working with incomplete or obscured evidence, since real outcrops may only show part of a cross-set, have a highly weathered surface, or present an imperfect version of a structure.

### 5.1 Discussion of bedding style and sedimentary structures

To accurately describe bedding geometry and other sedimentary structures, it is important to examine multiple scales and orientations of the outcrop of interest. In the VFT, we achieved this through using outcrop photos and 3D outcrop models that could be rotated and zoomed in on, in conjunction with instructor guidance in annotating images virtually on Zoom (e.g., outlining key geometries), as one would point out key features in the field.

The types of bedding at Grand Ledge are limited and often highly weathered or vegetated, and students did not always pay close enough attention to distinguish small variations. Interestingly, in some relatively monotonous outcrops, students simply did not notice major sedimentary structures (such as large-scale cross-bedding) until an instructor guided their observations. Encouraging students to "zoom out" and examine the whole outcrop did help them to see all key sedimentary structures and bedding trends, but required instructor guidance. This is not dissimilar to in-person field experiences, in which students often focus on weathering features or fractures instead of the primary structures. An important aspect of preparation for the VFT is to show examples in previous classes of large-scale geometries, such as channel lenses.

In some outcrops, students took the initiative to zoom in as much as possible, and also utilize the imagery from their lithology investigations, and identified mm- or cm-scale features, such as flaser and lenticular bedding or small pockets of coal. This level of detail allowed students to confirm or revise their environmental interpretations with a high degree of confidence. Like an in-person field experience, the students who explored the entirety of an outcrop at all scales collected enough data to make the most accurate interpretations.

## 6 Stratigraphy: Part 5 of assignment





Part 5 of the VFT was to create a detailed, stratigraphic column using data acquired in previous sections (SLO #2-4) in order to
develop an interpretation of the depositional environment(s). Detailed instructions on how to construct a stratigraphic column as
well as several examples were provided for the students. Our instructions were adapted from exercises on constructing virtual
graphic logs (Bristow, 2020) and clastic facies analyses (Anderton, 1985). Our examples came from *Geological Field Techniques*
(Coe, 2010), the *Art of Geological Field Sketches* (Noad, 2016), *Geology Drawing Skills Handbook* (2018), various notes on field
geology and stratigraphy techniques (e.g. Dr. Susan Kidwell, personal communication, 2014), and our personal field books. A pdf
of a blank logging sheet was provided to the students to construct their log digitally (e.g. in a pdf viewer, Word, PowerPoint, or
Google slides) or on paper by hand in a standardized format.

Prior to the VFT, students prepared by learning how to make a stratigraphic column. In the online-only format at Albion, students
worked through a virtual graphic logs lab exercise (Bristow, 2020), preparing students to also draw using their computers. At
Calvin, students practiced through making an in-person outdoor stratigraphic column of campus buildings, and using supplements
from the VFT.

It was important to have all students use the same increments in their stratigraphic column for consistency. This allowed us to
compare and correlate the columns in Part 6. They were instructed to include the data listed in Table 1 to construct their stratigraphic
columns, drafting both a graphic log and including written notes on their data and interpretations. Students were instructed to make
interpretations and concise notes on their column, including hypotheses about paleoenvironmental interpretations and notes on
relative sea level change. Students were encouraged to continually revise their interpretations. This step of the VFT yielded
products of stratigraphic columns, which were explicitly to be their most refined product, an example of which is provided in
Figure 6. The construction of stratigraphic columns prepared students for a discussion and comparison of their data with their
peers. In the final stage of this part, students worked with their partner(s) to assemble a short set of Google Slides that included a
map showing their locality, an outcrop sketch, a stratigraphic column, and interpretations, to be presented in the next step of the
VFT.

| Key features to include for the VFT stratigraphic column: |
| --- |
| Thicknesses of beds and sediment geometry |
| Lithologies and texture information: Lithology is indicated by a pattern or a note beside the column. Grain size is indicated by the width along the X-axis, expressed with a ragged or smoothed edge as it changes |
| Sedimentary structures (physical and biogenic): Include symbols on the column and describe them at the side. |
| Fossil content and clasts: Include symbols on the column and describe them at the side. |
| Nature of contacts (sharp? erosional features? relief?) |
| Weathering style of individual beds (note changes in color and if a unit is more recessive vs. more ledgy or resistant) |

**Table 1:** Key features to include for drafting a stratigraphic column in the Grand Ledge VFT.



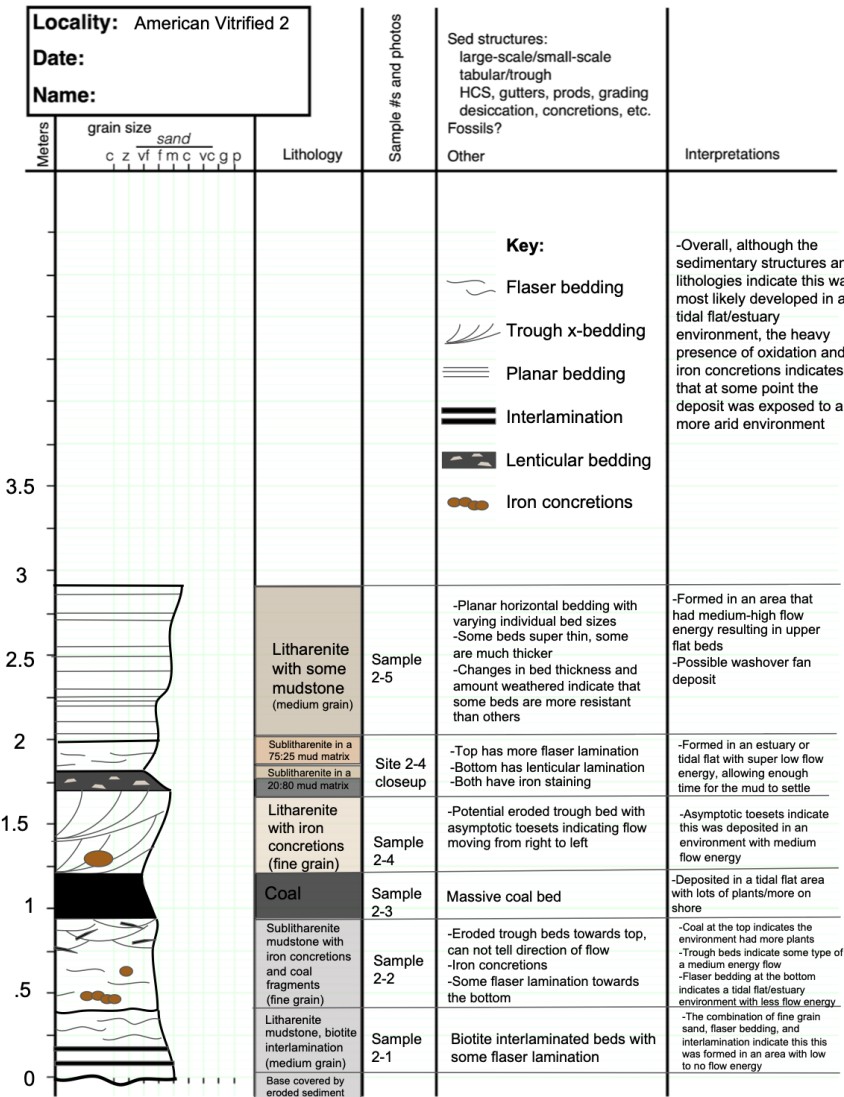


**Figure 6:** Example student stratigraphic section drafted from American Vitrified 2 outcrop data. This is a product from Part 5 of the Assignment, using data collected in Parts 1-4 of the Assignment (examples of data shown in Figs. 1-5). Stratigraphic columns that students produce include the graphic log, descriptions, and interpretations.

### 6.1 Discussion of stratigraphy

To compile a stratigraphic column with sufficient detail and precision, it was important for students to incorporate their observations from the previous sections of the VFT in an organized manner, using a standardized logging sheet and symbols. The Albion students completed all of their stratigraphic sections digitally, annotating the log file in pdf viewing software, or inserting the .png file into Word, PowerPoint, or a Google Slide to draw upon. One consideration is that it took students a long time to construct columns digitally due to the challenge of manipulating a laptop trackpad. By typing descriptions and interpretations,

students were able to include more details than when handwriting on the same logging sheet. Calvin students drafted theirs on paper by hand (Figure 7 shows examples of their final products). The problems students encountered in drafting their stratigraphic



columns were largely similar for both groups of students, though none were significant. We describe some of the challenges and successes below.

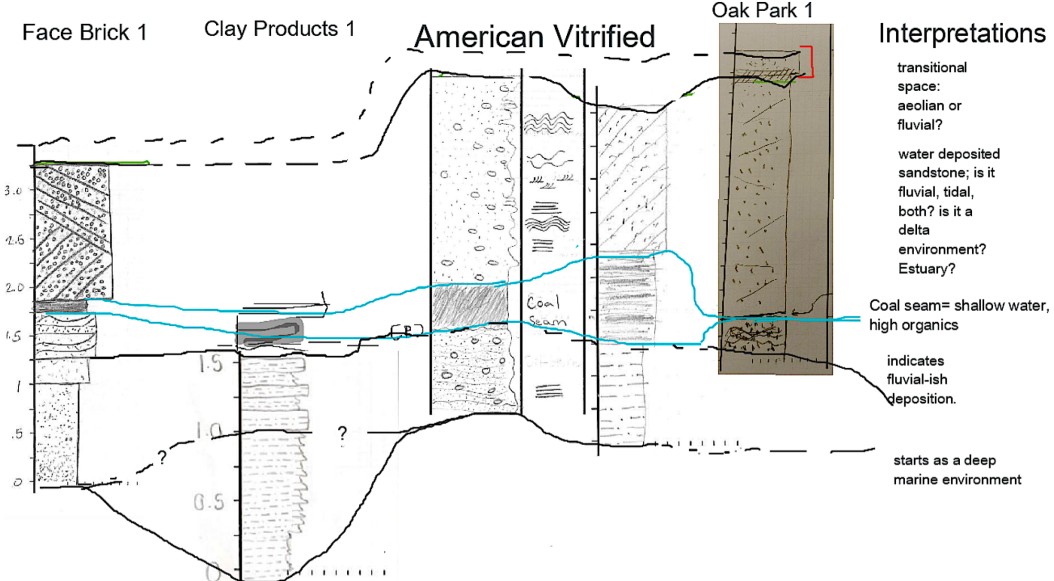

**Figure 7:** Example student correlation constructed in Part 7 of the Assignment by one class group using Google Jamboard, including hand-drawn representative stratigraphic sections drafted in Parts 5-6.


Stratigraphic columns are essentially graphs of data, and yet students often took artistic liberties, resulting in fanciful bedding

scales, and unrealistic distributions of features. Students found it challenging to draw realistic representations of sedimentary structures, such as drawing the larger-scale cross-beds that included multiple sets; smaller sedimentary structures such as laminations and fossils were represented more successfully. They also found it challenging to represent the scope of outcrop features accurately. For example, students noticed the iron concretions in one unit and incorporated them accurately in their outcrop sketches, but then included the concretions throughout the entirety of the stratigraphic column, instead of only in the units in which

they occurred.

The blank lithologic log provided to students for drafting their sections did not include a numbered scale. Part of the instructions guided students through how and where to number their scale, but this was not foolproof. A future modification could include a numerical scale already on the log, which will save time and prevent scaling errors between students that hinder correlations with

other sections. Interpretations of the columns yielded inconsistent results: students varied in their responses, with some simply reiterating their descriptions, others giving a broad interpretation for the entire column, and some detailing a paleoenvironmental interpretation for each unit (the goal). Students were encouraged to create multiple drafts, in order to yield a professional final product.

Most students displayed all geologic units accurately and consistently, incorporating both patterns and descriptions of lithologies. Many sections included a sufficient level of detail, particularly in the lithologic patterns and notes of other features, such as fossils,





bedding, and concretions. While students did include joints and fractures in their realistic outcrop sketches, they successfully excluded those secondary features from their stratigraphic columns.

## 7 Collaborate: Part 6 of assignment

In the previous part of the VFT, each student drafted a stratigraphic column for their assigned outcrop (SLO #4; Fig. 6). In Part 6, students collaborated with class members to discuss and revise their interpretations of the depositional environment(s) for their stratigraphic columns (SLO #5). The groups met with 1-2 other groups (now a "pod") working at adjacent outcrop localities to collaborate and share ideas. To start, each outcrop group presented an overview of their outcrop data and interpretations to the pod. To aid their discussion, the pod organized representative stratigraphic columns from each outcrop into one document (we used

Google Jamboard, a collaborative online whiteboard), retaining scale across the sites. With the outcrops in place for comparison, the groups studied the sites for similarities and differences. Using Google Jamboard, they began making correlations between outcrops, tracing marker beds or distinctive contacts or facies changes. To support the graphical correlations, students also added text to their Jamboard, describing particular similarities, differences, and uncertainties across the sections (see Figure S1). Additionally, students compared and contrasted individual environmental interpretations, and then worked as a pod to refine a

written paleoenvironmental history for their general area.

### 7.1 Discussion of collaboration

Students were successful in this part of the VFT in seeing how adjacent stratigraphy could be similar or different, and how stratigraphic sections could consistently be correlated with key features. For example, students recognized a coal seam regionally, and noted that it was in most sections – this provided a point of familiarity and drove their correlation decisions. This was the part

of the project where students had big "lightbulb" moments of understanding the connections. For the first time, students were learning how to relate one stratigraphic column to another.

Uneven amounts of participation and commitment from the small groups within a pod was a challenge. This could potentially be ameliorated by providing open access to all groups' stratigraphic columns and interpretations. Some groups had a very good

collaborative dynamic, while others did not. This is often challenging to predict in advance, whether students choose their own partners or are assigned groups. This was the most crucial part of the VFT for having developed functional group dynamics with initial partners. Jigsaw activities and group projects are contingent on all students being present and participating. When any student is absent, this presents challenges for the entire group. We attempted to address this challenge by having pods of three small groups, so that if one group did not engage, the remaining two groups could still collaborate. More importantly, for heading into Part 7, it

was important that all students felt comfortable with the data for their outcrops and adjacent outcrops.

Ideally, students would return to their stratigraphic columns at this point to correct misconceptions or errors in sketching their observations that they discovered upon discussion and peer review with the larger pod. A significant challenge is getting students to ask questions of each other or of their own work. In the future, it would be beneficial for students to be provided with examples

of useful questions to ask. As instructors monitor the pods, they could explicitly model the types of questions to ask and how to modify stratigraphic columns.



For time constraints, it is possible to proceed directly to the full-class Jamboard (Part 7) from individual sections (Part 5), skipping Part 6. However, the key realization that different outcrops even within close proximity to each other can show very different

features, or help to fill in gaps and confusion before doing a larger-scale correlation (addressed in Part 6), is a useful step. Practice presenting and organizing thoughts in a pod setting prior to full-class presentations was also important.

## 8 Disseminate knowledge:  Part 7 of assignment

Part 7 of the VFT asked students to bring the previous stages of the project together, collaborating as a full class to develop a correlation and an interpretation of the range of depositional environments for the entire field area, and then writing final reflections

(SLO #6). Each small group was tasked with selecting a representative stratigraphic column from their site and presenting the columns and their preliminary interpretations to the full class. The pods of small groups with adjacent outcrops were also given the opportunity to present and justify their preliminary correlations. These virtual presentations were done using either Google Slides or Google Jamboard. Next, each small group added their representative stratigraphic column to a full-class Jamboard. The class used the live-annotate functions of the Jamboard to virtually draft correlations between sections, debated lateral relationships,

and drew upon their knowledge of similar sedimentary records from readings and class to generate a cohesive set of hypotheses about the depositional system (Fig. 7). To equip students to make more explicit environmental interpretations, two additional readings were assigned, which describe modern examples of tidal inlets and washover fans (Pierce, 1970), and discriminate between tidal versus fluvial influences on sedimentation (Johnson and Dashtgard, 2014).

Following this work of sharing data and interpretations, students read a short field guide to the Grand Ledge area (Milstein, 1987a), which presents a clear interpretation of the site for students to evaluate. To conclude the VFT, students wrote a final reflection summarizing what they learned, and discussing how the final class hypotheses and interpretations compared with their original hypotheses and with published interpretations. In addition, students reflected on what went well and what to change about the experience.

**8.1 Discussion of dissemination of knowledge**

Part 7 of the project took a big-picture perspective, bringing all previous parts together scientifically, while also asking students to present their ideas and reflect on their learning. Students were enthusiastic as they saw how everything fit together. They felt like "real geologists" in this part, and enjoyed seeing how all the data connected and explained a real-world scenario. Students improved their skills at graphically representing ideas by applying their knowledge from the collaboration in Part 6 to the bigger picture.


It is important to remind students to draw upon the multiple scales of data they worked with throughout the VFT. This leads to more nuanced hypotheses and interpretations, by incorporating all available data from different perspectives:  for example, sometimes the best evidence to support a final correlation hypothesis was from the thin sections, which were studied in Part 3. When it came to interpreting paleoenvironments, it was important to remind students to zoom out and recognize the lateral variation

of landscapes, and consider the expected lateral and vertical successions of facies.

It was challenging to ensure that all students contributed to the correlation, since the most confident students often take the lead and complete this work first. Some students struggled to incorporate all their data and observations into a final interpretation,



relying heavily on only one or two parts of the project for their final reflections, instead of uniting the sedimentology, stratigraphy,
correlations, and related readings.

Since this VFT required teamwork, a useful component to both evaluating student work and giving students a voice in their
outcomes was to collect peer and self-evaluation reflections. Students were asked to numerically assess their own and their partners'
performance on a scale of 1-5, and also to provide qualitative assessment on how effectively their group worked together and any
behaviors of team members that were particularly valuable or detrimental to the team. This information was used to assess student
performance and classroom dynamics, but has no bearing on our evaluation of the VFT as presented in this study.

## 9 Summary of student feedback

Table S1 summarizes a variety of student reflections related to the Student Learning Outcomes and their experiences with the VFT
overall. We focused on feedback related to why students did or did not like the VFT, what was challenging and how challenges
were overcome, and achievement of the project SLOs. Prominent themes from the student's final reflections include:

1. Students gained a sense for the challenges associated with all aspects of fieldwork. For example, they identified note-taking as an aspect of fieldwork they had not previously realized was important or difficult.
2. Students appreciated the opportunity to apply what they had learned in class to a real-world, imperfect situation with natural variation.
3. Students reported that they learned how to form hypotheses about sedimentary environments, and revise their hypotheses based on additional information and collaboration.
4. Students noted how important collaboration was in solving problems, and that teamwork was useful even when challenging.

## 10 Conclusions

This VFT about Grand Ledge, Michigan provided an opportunity for a remote field experience for two Sedimentation and
Stratigraphy classes at liberal arts colleges. The success of this VFT depended on all components being prepared in advance and
carefully structured. This created continuity and a guided experience, while still allowing students the freedom to explore within
each part of the VFT. Through this VFT, students were able to access more data than during a traditional field trip (i.e., thin
sections), leading to a complete experience in which they could develop hypotheses and also access the multiple types of data
needed to test the hypotheses and refine their interpretations. This VFT facilitated a learning process that asked students to utilize
and build upon multiple skill sets and content areas in a single continuous framework, which students perceived as important for
developing their skills as geologists. The student products and feedback suggest that the VFT can yield the benefits of a traditional
field experience. The products, such as student field notes, stratigraphic columns, and group interpretations support this, as well as
the prominent themes from the student feedback. Therefore, the VFT is viable if used independently or could be used in conjunction
with an in-person field trip. Importantly, this VFT provided students with a realistic experience related to a local field area and all
of its natural variations and complexities, all within an accessible format and achievable within designated class time.

## Data availability



Access to the collection of Grand Ledge virtual 3D models of outcrops and hand samples created by Madeline Marshall and used in this VFT is available via Sketchfab at https://skfb.ly/6ZTzo (in the collections of albionsedpaleo).

All other VFT data and materials are available at
https://serc.carleton.edu/NAGTWorkshops/online_field/activities/242310.html.

**Supplement**

The supplement related to this article is available online at:

**Author contributions**

MM developed the VFT concept; MM and MH collected the VFT data, designed, and co-wrote the paper.

**Competing interests**

The authors declare that they have no conflict of interest.

**Acknowledgements**

We would like to thank our students at both Albion and Calvin; the 2020 NAGT Early Career Workshop; Ralph Stearley, John
VanRegenmorter, Ian Winkelstern for fieldwork discussions and assistance; and Crystal Bruxvoort and William Bartels for helpful suggestions and feedback on our manuscript.

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
