# Peer review of "Multi-Scale Virtual Field Experience: Sedimentology and stratigraphy of Grand Ledge, Michigan, USA"

_Geoscience Communication, 2021_

## Referee Comment (RC2)

**Review by Kent M. Syverson, Dept. of Geology, University of Wisconsin-Eau Claire**

**29 June 2021**

**General Comments** for "*Multi-scale virtual field experience, Grand Ledge, Michigan, USA*" a manuscript written by M.S. Marshall and M.C. Higley, gc-2021-10

The manuscript by Marshall and Higley describes a virtual field trip (VFT) for a classic sed/strat field trip site, Grand Ledge, in Michigan, USA. Given COVID and efforts to make geology more accessible to individuals who might have a physical disability, bringing field-based exercises "out of the field and into the lab or home" has become a critical issue. The work is especially appropriate for Copernicus special issue on Virtual geoscience education resources. I suggest modifying the title to include the words "sedimentology and stratigraphy" to catch the eyes of university sed/stratigraphy instructors.

The VFT contains materials intended to simulate an in-person visit to the sedimentary rock outcrops at Grand Ledge. The authors have used different types of technology to create 3D outcrop and hand sample images, photographs of different parts of the outcrops with multiple scales, and photomicrographs referenced to hand samples and the larger outcrops. These technologies are not new, but seeing these technologies applied to an exercise for undergraduate students is novel. In addition, the authors walk the reader through the entire exercise and explain student successes and challenges while working on the exercise. As a package, this VFT discussion is worthy of publication.

I have examined the article and the supplemental materials provided by the authors. I am impressed by the amount of work invested in this project and think it is useful to other geology instructors. In viewing the 3D models of the outcrops and hand samples, I would like to see more up/down reference points in the images. When I am in the field, I carefully observe the original orientations of samples. For an unexperienced student, I suspect they could literally "get turned around" as they rotate samples and outcrops in the 3D views. In addition, I wonder if the VFT might become less useful in several years after many cohorts of students are looking at the same images (work gets passed on to future students?).

**Specific Comments**

    1. **Does the paper address relevant scientific questions within the scope of GC?**

Yes. The work is especially appropriate for Copernicus special issue on Virtual geoscience education resources.

    2. **Does the paper present novel concepts, ideas, tools, or data?**

The authors have used different types of technology to create 3D outcrop and hand sample images, photographs of different parts of the outcrops with multiple scales, and photomicrographs referenced to hand samples and the larger outcrops. These technologies are not new, but seeing these technologies applied to an exercise for undergraduate students is novel and extremely useful as we prepare for the post-COVID world, the next pandemic, and making geology sites available to people who might have physical disabilities.

3. **Are the scientific methods and assumptions valid and clearly outlined?**

Yes.

4. **Are the results sufficient to support the interpretations and conclusions?**

Yes. The authors do not pretend to show in a quantitative way that their VFT is effective. However, they do offer student comments suggesting students have had a valuable educational experience.

5. **Do the authors give proper credit to related work and clearly indicate their own new/original contribution?**

Yes.

6. **Does the title clearly reflect the contents of the paper?**

I think the current title, "***Multi-scale virtual field experience, Grand Ledge, Michigan, USA***," could be improved.

This article would be of greatest interest to a teacher of sedimentology and stratigraphy in a college/university setting. I think the authors should somehow mention sed/strat in their title to attract the appropriate audience. Suggestions:

*Multi-scale virtual field experience for sedimentology/stratigraphy, Grand Ledge, Michigan, USA*

*Virtual field experience for a well-known sedimentology/stratigraphy site, Grand Ledge, Michigan, USA.*

7. **Does the abstract provide a concise and complete summary?**

Yes.

8. **Is the overall presentation well structured and clear?**

Yes.

9. **Is the language fluent and precise?**

In general, yes.  However, in some cases the authors overuse personal pronouns, and this makes the text overly wordy in places.  In other places lists do not follow rules of parallel construction.  In addition, the authors should search for the word "that" and evaluate if the word is necessary or can be removed by rewording the sentence.  Some specific areas:

Line 43—wordy

Line 40-47 – parallel construction for last item in their list

Line 67 –  Wordy -- "First, they submitted  notes,".  This is just one example.  The authors could delete personal pronoun elsewhere to tighten the text.

Line 150 – verb tense

Line 341 – ambiguous – reword

Line 340-345 – unclear

Line 364 – parallel construction with verb tense

**10. Are the number and quality of references appropriate?**

Yes.  I am not an expert in geoscience educational research, but the references cited seem recent and meaningful.

---

## Author Response (AR2)

**Multi-Scale Virtual Field Experience: Sedimentology and stratigraphy of Grand Ledge, Michigan, USA**

Madeline S. Marshall[1], Melinda C. Higley[2]

[1]Department of Earth & Environment, Albion College, Albion, Michigan, 49224, USA
[2]Geology, Geography, and Environment Department, Calvin University, Grand Rapids, Michigan, 49546, USA

*Correspondence to*: Madeline S. Marshall (mmarshall@albion.edu)

**Revisions and Edits**

Line numbers refer to review preprint; our revisions address comments from reviewers RC1 and RC2.

**Line 1:** We revised the title to:  Multi-Scale Virtual Field Experience, Sedimentology and stratigraphy of Grand Ledge, Michigan, USA

**Line 43:** To address wordiness, we replaced the phrase "(2) creating a structure that would be expandable and ongoing in its scope, with our future goals being to incorporate subsurface data and samples from the Michigan Core Repository; " with "(2) creating an expandable structure, with future goals to incorporate subsurface data and samples from the Michigan Core Repository;"

**Line 44:**  We replaced the phrase "inflexible scope" with  "limited scales of data, and inflexible implementation."

**Line 40-47:** We fixed the parallel construction for last item in the list of this paragraph by replacing "(4) a broader goal of this Grand Ledge VFT was to thoroughly document and encourage the preservation of a suite of historically and geologically important Pennsylvanian outcrops in Grand Ledge, Michigan (e.g., Milstein, 1987a)." with "(4) thoroughly documenting to encourage the preservation of a suite of historically and geologically important Pennsylvanian outcrops in Grand Ledge, Michigan (e.g., Milstein, 1987a)."

**Line 67:**  To address wordiness, we replaced the phrase "First, they submitted a copy of their field notes," with "first they submitted field notes".

**Line 90:** We added a sentence to explain that the panorama incurred distortion during the photomerging process. We chose to mention this in section 1.3 VFT Development to account for all panoramas in the project, not just the one shown in Figure 1.

**Line 150**: We rephrased this sentence to clarify it was comparing in-person to virtual fieldwork during this stage of the experience.

**Line 151:** Deleted an unnecessary "that."

**Line 172:** Deleted an unnecessary "that."

**Line 341**: This sentence was ambiguous and was deleted: "This was the most crucial part of the VFT for having developed functional group dynamics with initial partners."

**Line 340-345**: We rephrased this section to clarify our meaning.

**Line 364:** We fixed the parallel construction by replacing "The class used the live-annotate functions of the Jamboard to virtually draft correlations between sections, debated lateral relationships, and drew upon their knowledge of similar sedimentary records from readings and class to generate a cohesive set of hypotheses about the depositional system (Fig. 7). " with "The class used the live-annotate functions of the Jamboard to virtually draft correlations between sections, **debate** lateral relationships, and **draw** upon their knowledge of similar sedimentary records from readings and class to generate a cohesive set of hypotheses about the depositional system (Fig. 7). "

**Line 398:** We clarified how student responses were coded and processed by including the following at this point: "In reading student reflections, we selected representative student responses about their learning outcomes that corresponded to each SLO, and coded these responses by SLO and school to ensure an even distribution of comments from Albion and Calvin students. Student experiences and outcomes were markedly similar between the two schools, and comments from reflections were aggregated. We excluded student reflection responses that were not related to an SLO, or were focused on specifics they learned about their particular outcrop. Student reflections were intended to be open-ended, as opposed to a structured survey; thus, further qualitative data analysis is beyond the scope of this project."

**Line 399-400:** We moved the following sentence to be placed appropriately within the newly added explanation of how we used student responses: "We focused on feedback related to why students did or did not like the VFT, what was challenging and how challenges were overcome, and achievement of the project SLOs."